# Backpropagation-Friendly Eigendecomposition

Wei Wang[1], Zheng Dang[2], Yinlin Hu[1], Pascal Fua[1], and Mathieu Salzmann[1]

[1]CVLab, EPFL, CH-1015 Lausanne, Switzerland {first.last}@epfl.ch
[2]Xi'an Jiaotong University, China {dangzheng713@stu.xjtu.edu.cn}

## Abstract

Eigendecomposition (ED) is widely used in deep networks. However, the backpropagation of its results tends to be numerically unstable, whether using ED directly or approximating it with the Power Iteration method, particularly when dealing with large matrices. While this can be mitigated by partitioning the data in small and arbitrary groups, doing so has no theoretical basis and makes its impossible to exploit the power of ED to the full.

In this paper, we introduce a numerically stable and differentiable approach to leveraging eigenvectors in deep networks. It can handle large matrices without requiring to split them. We demonstrate the better robustness of our approach over standard ED and PI for ZCA whitening, an alternative to batch normalization, and for PCA denoising, which we introduce as a new normalization strategy for deep networks, aiming to further denoise the network's features.

## 1 Introduction

In recent years, Eigendecomposition (ED) has been incorporated in deep learning algorithms to perform PCA whitening [1], ZCA whitening [2], second-order pooling [3], generative modeling [4, 5], keypoint detection and matching [6–8], pose estimation [9], camera self-calibration [10], and graph matching [11]. This requires backpropagating the loss through the ED, which can be done but is often unstable. This is because, as shown in [3], the partial derivatives of an ED-based loss depend on a matrix $\widetilde{\mathbf{K}}$ with elements

$$\widetilde{K}_{ij} = \left\{ \begin{array}{ll} 1/(\lambda_i - \lambda_j), & i \neq j \\ 0, & i = j \end{array} \right. , \tag{1}$$

where $\lambda_i$ denotes the $i^{\text{th}}$ eigenvalue of the matrix being processed. Thus, when two eigenvalues are very close, the partial derivatives become very large, causing arithmetic overflow.

The Power Iteration (PI) method [12] is one way around this problem. In its standard form, PI relies on an iterative procedure to approximate the dominant eigenvector of a matrix starting from an initial estimate of this vector. This has been successfully used for graph matching [11] and spectral normalization in generative models [4], which only require the largest eigenvalue. In theory, PI can be used in conjunction with a *deflation* procedure [13] to find *all* eigenvalues and eigenvectors. This involves computing the dominant eigenvector, removing the projection of the input matrix on this vector, and iterating. Unfortunately, two eigenvalues being close to each other or one being close to zero can trigger large round-off errors that eventually accumulate and result in inaccurate eigenvector estimates. The results are also sensitive to the number of iterations and to how the vector is initialized at the start of each deflation step. Finally, the convergence speed decreases significantly when the ratio between the dominant eigenvalue and others becomes close to one.

In short, both SVD [3] and PI [11] are unsuitable for use in a deep network that requires the computation of gradients to perform back-propagation. This is particularly true when dealing with large matrices for which the chances of two eigenvalues being almost equal is larger than for small

ones. This why another popular way to get around these difficulties is to use smaller matrices, for example by splitting the feature channels into smaller groups before computing covariance matrices, as in [2] for ZCA whitening [14, 15]. This, however, imposes arbitrary constraints on learning. They are not theoretically justified and may degrade performance.

In this paper, we therefore introduce a numerically stable and differentiable approach to performing ED within deep networks in such as way that their training is robust. To this end, we leverage the fact that the forward pass of ED is stable and yields accurate eigenvector estimates. As the aforementioned problems come from the backward pass, once the forward pass is complete we rely on PI for backprogation purposes, leveraging the ED results for initialization. We will show that our hybrid training procedure consistently outperforms both ED and PI in terms of stability for

- ZCA whitening [2]: An alternative to Batch Normalization [16], which involves linearly transforming the feature vectors so that their covariance matrices becomes the identity matrix and that they can be considered as decoupled.
- PCA denoising [17]: A transformation of the data that reduces the dimensionality of the input features by projecting them on a subspace spanned by a subset of the eigenvectors of their covariance matrix, and projects them back to the original space. Here, we introduce PCA denoising as a new normalization strategy for deep networks, aiming to remove the irrelevant signal from their feature maps.

Exploiting the full power of both these techniques requires performing ED on relatively large matrices and back-propagating the results, which our technique allows whereas competing ones tend to fail. The code is available at `https://github.com/WeiWangTrento/Power-Iteration-SVD`.

## 2 Numerically Stable Differentiable Eigendecomposition

Given an input matrix $\mathbf{M}$, there are two standard ways to exploit the ED of $\mathbf{M}$ within a deep network:

1. Perform ED using SVD or QR decomposition and use analytical gradients for backpropagation.
2. Given *randomly-chosen* initial guesses for the eigenvectors, run a PI deflation procedure during the forward pass and compute the corresponding gradients for backpropagation purposes.

Unfortunately, as discussed above, the first option is prone to gradient explosion when two or more eigenvalues are close to each other while the accuracy of the second strongly depends on the initial vectors and on the number of iterations in each step of the deflation procedure. This can be problematic because the PI convergence rate depends geometrically on the ratio $|\lambda_2/\lambda_1|$ of the two largest eigenvalues. Therefore, when this ratio is close to 1, the eigenvector estimate may be inaccurate when performing a limited number of iterations. This can lead to training divergence and eventual failure, as we will see in the results section.

Our solution is to rely on the following hybrid strategy:

1. Use SVD during the forward pass because, by relying on a divide-and-conquer strategy, it tends to be numerically more stable than QR decomposition [12].
2. Compute the gradients for backpropagation from the PI derivations, but using the SVD-computed vectors for initialization purposes.

In the remainder of this section, we show that the resulting PI gradients not only converge to the analytical ED ones, but are bounded from above by a factor depending on the number of PI iterations, thus preventing their explosion in practice. In the results section, we will empirically confirm this by showing that training an ED-based deep network using our approach consistently converges, whereas the standard SVD and PI algorithms often diverge.

### 2.1 Power Iteration Gradients

Let $\mathbf{M}$ be a covariance matrix, and therefore be positive semi-definite and symmetric. We now focus on the leading eigenvector of $\mathbf{M}$. Since the deflation procedure simply iteratively removes the projection of $\mathbf{M}$ on its leading eigenvector, the following derivations remain valid at any step of this procedure. To compute the leading eigenvector $\mathbf{v}$ of $\mathbf{M}$, PI relies on the iterative update

$$\mathbf{v}^{(k)} = \mathbf{M}\mathbf{v}^{(k-1)}/\|\mathbf{M}\mathbf{v}^{(k-1)}\| \, , \tag{2}$$

where $\|\cdot\|$ denotes the $\ell_2$ norm. The PI gradients can then be computed as [18].

$$\frac{\partial L}{\partial \mathbf{M}} = \sum_{k=0}^{K-1} \frac{\left(\mathbf{I}-\mathbf{v}^{(k+1)}\mathbf{v}^{(k+1)\top}\right)}{\|\mathbf{M}\mathbf{v}^{(k)}\|} \frac{\partial L}{\partial \mathbf{v}^{(k+1)}} \mathbf{v}^{(k)\top} \;,\; \frac{\partial L}{\partial \mathbf{v}^{(k)}}=\mathbf{M}\frac{\left(\mathbf{I}-\mathbf{v}^{(k+1)}\mathbf{v}^{(k+1)\top}\right)}{\|\mathbf{M}\mathbf{v}^{(k)}\|} \frac{\partial L}{\partial \mathbf{v}^{(k+1)}} \;. \quad (3)$$

Typically, to initialize PI, $\mathbf{v}^{(0)}$ is taken as a random vector such that $\|\mathbf{v}^{(0)}\|=1$. Here, however, we rely on SVD to compute the true eigenvector $\mathbf{v}$. Because $\mathbf{v}$ is an accurate estimate of the eigenvector, feeding it as initial value in PI will yield $\mathbf{v}=\mathbf{v}^{(0)}\approx\mathbf{v}^{(1)}\approx\mathbf{v}^{(2)}\approx\cdots\approx\mathbf{v}^{(k)}\cdots\approx\mathbf{v}^{(K)}$. Exploiting this in Eq. 3 and introducing the explicit form of $\frac{\partial L}{\partial \mathbf{v}^{(k)}}$, $k=1,2,\cdots,K$, into $\frac{\partial L}{\partial \mathbf{M}}$ lets us write

$$\frac{\partial L}{\partial \mathbf{M}} = \left( \frac{\left(\mathbf{I} - \mathbf{v}\mathbf{v}^\top\right)}{\|\mathbf{M}\mathbf{v}\|} + \frac{\mathbf{M}\left(\mathbf{I} - \mathbf{v}\mathbf{v}^\top\right)}{\|\mathbf{M}\mathbf{v}\|^2} + \cdots + \frac{\mathbf{M}^{K-1}\left(\mathbf{I} - \mathbf{v}\mathbf{v}^\top\right)}{\|\mathbf{M}\mathbf{v}\|^K} \right) \frac{\partial L}{\partial \mathbf{v}^{(K)}} \mathbf{v}^\top \;. \quad (4)$$

The details of this derivation are provided in the supplementary material. In our experiments, Eq. 4 is the form we adopt to compute the ED gradients.

## 2.2 Relationship between PI and Analytical ED Gradients

We now show that when $K$ goes to infinity, the PI gradients of Eq. 4 are the same as the analytical ED ones.

**Power Iteration Gradients Revisited.** To reformulate the PI gradients, we rely on the fact that

$$\mathbf{M}^k = \mathbf{V}\mathbf{\Sigma}^k\mathbf{V}^\top = \lambda_1^k\mathbf{v}_1\mathbf{v}_1^\top + \lambda_2^k\mathbf{v}_2\mathbf{v}_2^\top + \cdots + \lambda_n^k\mathbf{v}_n\mathbf{v}_n^\top \;, \quad (5)$$

and that $\|\mathbf{M}\mathbf{v}\| = \|\lambda\mathbf{v}\| = \lambda$, where $\mathbf{v} = \mathbf{v}_1$ is the dominant eigenvector and $\lambda = \lambda_1$ is the dominant eigenvalue. Introducing Eq. 5 into Eq. 4, lets us re-write the gradient as

$$\begin{aligned}
\frac{\partial L}{\partial \mathbf{M}} &= \left( \frac{\left(\sum_{i=2}^n \mathbf{v}_i\mathbf{v}_i^\top\right)}{\lambda_1} + \frac{\left(\sum_{i=2}^n \lambda_i\mathbf{v}_i\mathbf{v}_i^\top\right)}{\lambda_1^2} + \cdots + \frac{\left(\sum_{i=2}^n \lambda_i^{K-1}\mathbf{v}_i\mathbf{v}_i^\top\right)}{\lambda_1^K} \right) \frac{\partial L}{\partial \mathbf{v}_1^{(K)}} \mathbf{v}_1^\top \\
&= \left( \sum_{i=2}^n \left( \frac{1}{\lambda_1} + \frac{1}{\lambda_1}\left(\lambda_i/\lambda_1\right)^1 + \cdots + \frac{1}{\lambda_1}\left(\lambda_i/\lambda_1\right)^{K-1} \right) \mathbf{v}_i\mathbf{v}_i^\top \right) \partial L/\partial\mathbf{v}_1^{(K)} \mathbf{v}_1^\top \;.
\end{aligned} \quad (6)$$

Eq. 6 defines a geometric progression. Given that

$$1 - \left(\lambda_i/\lambda_1\right)^k \to 1, \;\text{ when } k \to \infty, \;\text{ because } |\lambda_i/\lambda_1| \leq 1,$$

we have

$$\frac{1}{\lambda_1} + \frac{1}{\lambda_1}\left(\frac{\lambda_i}{\lambda_1}\right)^1 + \cdots + \frac{1}{\lambda_1}\left(\frac{\lambda_i}{\lambda_1}\right)^{k-1} = \frac{\frac{1}{\lambda_1}\left(1 - \left(\frac{\lambda_i}{\lambda_1}\right)^k\right)}{1 - \frac{\lambda_i}{\lambda_1}} \to \frac{\frac{1}{\lambda_1}}{1 - \frac{\lambda_i}{\lambda_1}}, \text{ when } k \to \infty. \quad (7)$$

Introducing Eq. 7 into Eq. 6 yields

$$\frac{\partial L}{\partial \mathbf{M}} = \left( \sum_{i=2}^n \left( \frac{\frac{1}{\lambda_1}}{1 - \frac{\lambda_i}{\lambda_1}} \right) \mathbf{v}_i\mathbf{v}_i^\top \right) \frac{\partial L}{\partial \mathbf{v}_1^{(k)}} \mathbf{v}_1^\top = \left( \sum_{i=2}^n \frac{\mathbf{v}_i\mathbf{v}_i^\top}{\lambda_1 - \lambda_i} \right) \frac{\partial L}{\partial \mathbf{v}_1^{(k)}} \mathbf{v}_1^\top \;. \quad (8)$$

**Analytical ED Gradients** As shown in [3], the analytic form of the ED gradients can be written as

$$\frac{\partial L}{\partial \mathbf{M}} = \mathbf{v}_1 \left\{ \left( \widetilde{K}^\top \circ \left( \mathbf{v}_1^\top \frac{\partial L}{\partial \mathbf{v}_1} \right) \right) + \left( \frac{\partial L}{\partial \mathbf{\Sigma}} \right)_{diag} \right\} \mathbf{v}_1^\top \;, \quad (9)$$

where $\widetilde{K}$ given by Eq. 1. By making use of the same properties as before, this can be re-written as

$$\frac{\partial L}{\partial \mathbf{M}} = \sum_{i=2}^n \frac{1}{\lambda_1 - \lambda_i}\mathbf{v}_i\mathbf{v}_i^\top \frac{\partial L}{\partial \mathbf{v}_1} \mathbf{v}_1^\top + \frac{\partial L}{\partial \lambda_i}\mathbf{v}_i\mathbf{v}_i^\top \;. \quad (10)$$

The the last term of Eq. 10 can be ignored because $\lambda_i$ is not involved in any computation during the forward pass. Instead, we compute the eigenvalue as the Rayleigh quotient $\mathbf{v}^\top\mathbf{M}\mathbf{v}/\mathbf{v}^\top\mathbf{v}$, which only depends on the eigenvector and thus only need the gradients w.r.t. to them. A detailed derivation is provided in the supplementary material.

This shows that the partial derivatives of $\mathbf{v}_1$ computed using PI have the same form as the analytical ED ones when $k \to \infty$. Similar derivations can be done for $\mathbf{v}_i, i = 2, 3, ....$. This justifies our use of PI to approximate the analytical ED gradients during backpropagation. We now turn to showing that the resulting gradient estimates are upper-bounded and can therefore not explode.

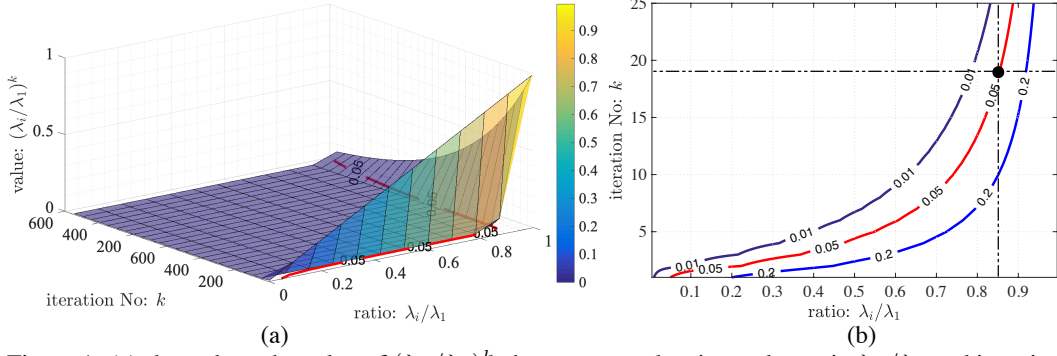

Figure 1: (a) shows how the value of $(\lambda_k/\lambda_1)^k$ changes *w.r.t.* the eigenvalue ratio $\lambda_k/\lambda_1$ and iteration number $k$. (b) shows the contour of curved surface in (a).

| $\lambda_i/\lambda_1$ | 0.1 | 0.2 | 0.3 | 0.4 | 0.5 | 0.6 | 0.7 | 0.8 | **0.85** | 0.9 | 0.95 | 0.99 | 0.995 | 0.999 |
|---|---|---|---|---|---|---|---|---|---|---|---|---|---|---|
| $k = \lceil \ln(0.05)/\ln(\lambda_i/\lambda_1) \rceil$ | 2 | 2 | 3 | 4 | 5 | 6 | 9 | 14 | **19** | 29 | 59 | 299 | 598 | 2995 |

Table 1: The minimum value of $k$ we need to guarantee $(\lambda_i/\lambda_1)^k < 0.05$.

## 2.3 Upper Bounding the PI Gradients

Recall that, when the input matrix has two equal eigenvalues, $\lambda_1 = \lambda_i$, the gradients computed from Eq.10 go to $\pm\infty$, as the denominator is 0. However, as Eq. 6 can be viewed as the geometric series expansion of Eq. 10, as shown below, it provides an upper bound on the gradient magnitude. Specifically, we can write

$$\frac{\partial L}{\partial \mathbf{M}} = \sum_{i=2}^{n} \frac{1}{\lambda_1 - \lambda_i} \mathbf{v}_i \mathbf{v}_i^\top \frac{\partial L}{\partial \mathbf{v}_1} \mathbf{v}_1^\top \approx \left( \sum_{i=2}^{n} \left( \frac{1}{\lambda_1} + \frac{1}{\lambda_1} q^1 \cdots + \frac{1}{\lambda_1} q^{K-1} \right) \mathbf{v}_i \mathbf{v}_i^\top \right) \frac{\partial L}{\partial \mathbf{v}_1} \mathbf{v}_1^\top , \quad (11)$$

where $q=\lambda_i/\lambda_1$. Therefore,

$$\left\| \frac{\partial L}{\partial \mathbf{M}} \right\| \leq \left\| \sum_{i=2}^{n} \left( \frac{1}{\lambda_1} + \frac{1}{\lambda_1} \left( \frac{\lambda_i}{\lambda_1} \right)^1 + \frac{1}{\lambda_1} \left( \frac{\lambda_i}{\lambda_1} \right)^2 + \cdots + \frac{1}{\lambda_1} \left( \frac{\lambda_i}{\lambda_1} \right)^{K-1} \right) \mathbf{v}_i \mathbf{v}_i^\top \right\| \left\| \frac{\partial L}{\partial \mathbf{v}_1} \right\| \|\mathbf{v}_1^\top\|$$

$$\leq \left\| \sum_{i=2}^{n} \left( \frac{1}{\lambda_1} \cdots + \frac{1}{\lambda_1} \right) \mathbf{v}_i \mathbf{v}_i^\top \right\| \left\| \frac{\partial L}{\partial \mathbf{v}_1} \right\| \|\mathbf{v}_1^\top\| \leq \sum_{i=2}^{n} \left\| \frac{K}{\lambda_1} \mathbf{v}_i \mathbf{v}_i^\top \right\| \left\| \frac{\partial L}{\partial \mathbf{v}_1} \right\| \|\mathbf{v}_1^\top\| \leq \frac{nK}{\lambda_1} \left\| \frac{\partial L}{\partial \mathbf{v}_1} \right\| . \quad (12)$$

This yields an upper bound of $\left\| \frac{\partial L}{\partial M} \right\|$. However, if $\lambda_1 = 0$, this upper bound also becomes $\infty$. To avoid this, knowing that $\mathbf{M}$ is symmetric positive semi-definite, we modify it as $\mathbf{M} = \mathbf{M} + \epsilon I$, where $I$ is the identity matrix. This guarantees that the eigenvalues of $\mathbf{M} + \epsilon I$ are greater than or equal to $\epsilon$. In practice, we set $\epsilon = 10^{-4}$. Thus, we can write

$$\left\| \frac{\partial L}{\partial (\mathbf{M} + \epsilon I)} \right\| \leq \frac{nK}{\epsilon} \left\| \frac{\partial L}{\partial \mathbf{v_1}} \right\| , \quad (13)$$

where $n$ is the dimension of the matrix and $K$ is the power iteration number, which means that choosing a specific value of $K$ amounts to choosing an upper bound for the gradient magnitudes. We now provide an empirical approach to doing so.

## 2.4 Choosing an Appropriate Power Iteration Number $K$

Recall from Eqs. 7 and 8 that, for the PI gradients to provide a good estimate of the analytical ones, we need $(\lambda_i/\lambda_1)^k$ to go to zero. Fig. 1 shows how the value $(\lambda_i/\lambda_1)^k$ evolves for different power iteration number $k$ and ratio $\lambda_i/\lambda_1$. This suggests the need to select an appropriate $k$ for each $\lambda_i/\lambda_1$.

To this end, we assume that $(\lambda_i/\lambda_1)^k \leq 0.05$ is a good approximation to $(\lambda_i/\lambda_1)^k = 0$. Then we have

$$(\lambda_i/\lambda_1)^k \leq 0.05 \Leftrightarrow k \ln(\lambda_i/\lambda_1) \leq \ln(0.05) \Leftrightarrow k \geq {\ln(0.05)}/{\ln(\lambda_i/\lambda_1)}. \quad (14)$$

That is, the minimum value of $k$ to satisfy $(\lambda_i/\lambda_1)^k \leq 0.05$ is $k = \lceil {\ln(0.05)}/{\ln(\lambda_i/\lambda_1)} \rceil$.

Table 1 shows the minimum number of iterations required to guarantee that our assumption holds for different values of $\lambda_i/\lambda_1$. Note that when two eigenvalues are very close to each other, *e.g.*, $\lambda_i/\lambda_1 = 0.999$, we need about 3000 iterations to achieve a good approximation. However, this is rare in practice. In CIFAR-10, using ResNet18, we observed that the average $\lambda_i/\lambda_{i-1}$ lies in $[0.84, 0.86]$

interval for the mini-batches. Given the values shown in Table 1, we therefore set the power iteration number to be $K = 19$, which yields a good approximation in all these cases.

## 2.5 Practical Considerations

In practice, we found that other numerical imprecisions due to the computing platform itself, such as the use of single vs double precision, could further affect training with eigendecomposition. Here, we discuss our practical approach to dealing with these issues.

The first issue we observed was that, in the forward pass, the eigenvalues computed using SVD may be inaccurate, with SVD sometimes crashing when the matrix is ill-conditioned. Recall that, to increase stability, we add a small value $\epsilon$ to the diagonal of the input matrix $\mathbf{M}$. As a consequence, all the eigenvalues should be greater than or equal to $\epsilon$. However, this is not the case when we use float instead of double precision. To solve this problem, we employ truncated SVD. That is, we discard the eigenvectors whose eigenvalue $\lambda_i \leq \epsilon$ and the subsequent ones.

The second issue is related to the precision of the eigenvalues computed using $\widetilde{\lambda}_i = \mathbf{v}^\top \widetilde{\mathbf{M}} \mathbf{v} / \mathbf{v}^\top \mathbf{v}$. Because of the round-off error from $\widetilde{\mathbf{M}} = \widetilde{\mathbf{M}} - \widetilde{\mathbf{M}} \mathbf{v}_i \mathbf{v}_i^\top$, $\widetilde{\lambda}_i$ may be inaccurate and sometimes negative. To avoid using incorrect eigenvalues, we also need to truncate it.

These two practical issues correspond to the breaking conditions $\lambda_i \leq \epsilon$, $\frac{\widetilde{\lambda}_i - \lambda_i}{\lambda_i} \geq 0.1$ defined in Alg.1 that provides the pseudo-code for our ZCA whitening application. Furthermore, we add another constraint, $\gamma_i \geq (1 - 0.0001)$, implying that we also truncate the eigendecomposition if the remaining energy in $\widetilde{\mathbf{M}}$ is less than 0.0001.

Note that this last condition can be modified by using a different threshold. To illustrate this, we therefore also perform experiments with a higher threshold, thus leading to our PCA denoising layer, which aims to discard the noise in the network features. As shown in our experiments, our PCA denoising layer achieves competitive performance compared with the ZCA one.

---

**Algorithm 1:** Forward Pass of ZCA whitening in Practice.

---

**Data:** $\mu = Avg(\mathbf{X})$,   $\widetilde{\mathbf{X}} = \mathbf{X} - \mu$,   $\mathbf{M} = \widetilde{\mathbf{X}} \widetilde{\mathbf{X}}^\top + \epsilon I$, ($\mathbf{X} \in R^{c \times n}$);
**Result:** $\mathbf{V}^\top \Lambda \mathbf{V} = SVD(\mathbf{M})$; $\Lambda = diag(\lambda_1, ..., \lambda_n)$; $\mathbf{V} = [\mathbf{v}_1, \cdots, \mathbf{v}_n]$; $\gamma_i = \sum_{k=1}^{i} \lambda_k / \sum_{k=1}^{n} \lambda_k$;
**Input:** $E_\mu \leftarrow 0$, $E_S \leftarrow I$, $\widetilde{\mathbf{M}} \leftarrow \mathbf{M}$, $rank \leftarrow 1$, $momentum \leftarrow 0.1$

1 **for** $i = 1 : n$ **do**
2     $\mathbf{v}_i$ = Power Iteration $(\widetilde{\mathbf{M}}, \mathbf{v}_i)$; $\widetilde{\lambda}_i = \mathbf{v}_i^\top \widetilde{\mathbf{M}} \mathbf{v}_i / (\mathbf{v}_i^\top \mathbf{v}_i)$; $\widetilde{\mathbf{M}} = \widetilde{\mathbf{M}} - \widetilde{\mathbf{M}} \mathbf{v}_i \mathbf{v}_i^\top$;
3     **if** $\lambda_i \leq \epsilon$  *or*  $|\widetilde{\lambda}_i - \lambda_i| / \lambda_i \geq 0.1$  *or*  $\gamma_i \geq (1 - 0.0001)$ **then**
4       break;
5     **else**
6       $rank = i$, $\widetilde{\Lambda} = [\widetilde{\lambda}_1, \cdots, \widetilde{\lambda}_i]$.
7     **end**
8 **end**
9 truncate eigenvector matrix: $\widetilde{\mathbf{V}} \leftarrow [\mathbf{v}_1, \cdots, \mathbf{v}_{rank}]$;
10 compute subspace: $\mathbf{S} \leftarrow \widetilde{\mathbf{V}}(\widetilde{\Lambda})^{-\frac{1}{2}} \widetilde{\mathbf{V}}^\top$;
11 compute ZCA transformation: $\mathbf{X} \leftarrow \mathbf{S} \widetilde{\mathbf{X}}$;
12 update running mean:     $E_\mu = momentum \cdot \mu + (1 - momentum) \cdot E_\mu$;
13 update running subspace: $E_{\mathbf{S}} = momentum \cdot \mathbf{S} + (1 - momentum) \cdot E_{\mathbf{S}}$;
**Output:** compute affine transformation: $\mathbf{X} = \gamma \mathbf{X} + \beta$

---

In Alg.1, the operation: $\mathbf{v}_i$ = Power Iteration$(\widetilde{\mathbf{M}}, \mathbf{v}_i)$ has no computation involved in the forward pass. It only serves to save $\widetilde{\mathbf{M}}, \mathbf{v}_i$ that will be used to compute the gradients during the backward pass. Furthermore, compared with standard batch normalization [16], we replace the running variance by a running subspace $E_{\mathbf{S}}$ but keep the learnable parameters $\gamma, \beta$. The running mean $E\mu$ and running subspace $E_{\mathbf{S}}$ are used in the testing phase.

## 3 Experiments

We now demonstrate the effectiveness of our Eigendecomposition layer by using it to perform ZCA whitening and PCA denoising. ZCA whitening has been shown to improve classification performance

| Methods | Error | Matrix Dimension | | | | |
|---|---|---|---|---|---|---|
| | | $d=4$ | $d=8$ | $d=16$ | $d=32$ | $d=64$ |
| SVD | Min | 4.59 | - | - | - | - |
| | Mean | **4.54 ± 0.08** | - | - | - | - |
| | Success Rate | 46.7% | 0% | 0% | 0% | 0% |
| PI | Min | 4.44 | 6.28 | - | - | - |
| | Mean | 4.99±0.51 | - | - | - | - |
| | Success Rate | **100%** | 6.7% | 0% | 0% | 0% |
| Ours | Min | 4.59 | 4.43 | **4.40** | 4.46 | 4.44 |
| | Mean | 4.71±0.11 | 4.62±0.18 | 4.63±0.14 | 4.64±0.15 | **4.59 ± 0.09** |
| | Success Rate | **100%** | **100%** | **100%** | **100%** | **100%** |

Table 2: Errors and success rates using ResNet 18 with standard SVD, Power Iteration (PI), and our method on CIFAR10. $d$ is the size of the feature groups we process individually.

| Methods | Error | BN | $d=64$ | $d=32$ | $d=16$ | $d=8$ | $d=4$ |
|---|---|---|---|---|---|---|---|
| ResNet18 | Min | 21.68 | 21.04 | 21.36 | 21.14 | 21.15 | **21.03** |
| | Mean | 21.85±0.14 | **21.39±0.23** | 21.58±0.27 | 21.45±0.25 | 21.56±0.35 | 21.51±0.28 |
| ResNet50 | Min | 20.79 | 19.28 | **19.24** | 19.78 | 20.15 | 20.66 |
| | Mean | 21.62±0.65 | 19.94±0.44 | **19.54±0.23** | 19.92±0.12 | 20.59±0.58 | 20.98±0.31 |

Table 3: Errors rates using ResNet18 or ResNet50 with Batch Normalization (BN) or our method on CIFAR100. $d$ is the size of the feature groups we process individually.

over batch normalization [2]. We will demonstrate that we can deliver a further boost by making it possible to handle larger covariance matrices within the network. PCA denoising has been used for image denoising [17] but not in a deep learning context, presumably because it requires performing ED on large matrices. We will show that our approach solves this problem.

In all the experiments below, we use either Resnet18 or Resnet50 [19] as our backbone. We retain their original architectures but introduce an additional layer between the first convolutional layer and the first pooling layer. For both ZCA and PCA, the new layer computes the covariance matrix of the feature vectors, eigendecomposes it, and uses the eigenvalues and eigenvectors as described below. As discussed in Section 2.4, we use $K = 19$ power iterations when backprogating the gradients unless otherwise specified. To accommodate this additional processing, we change the stride $s$ and kernel sizes in the subsequent blocks to Conv1(3×3,s=1)->Block1(s=1)->Block2(s=2)->Block3(s=2)->Block4(s=2)->AvgPool(4×4)->FC.

### 3.1 ZCA Whitening

The ZCA whitening algorithm takes as input a $d \times n$ matrix $\mathbf{X}$ where $d$ represents the feature dimensionality and $n$ the number of samples. It relies on eigenvalues and eigenvectors to compute a $d \times d$ matrix $\mathbf{S}$ such that the covariance matrix of $\mathbf{SX}$ is the $d \times d$ identity matrix, meaning that the transformed features are now decorrelated. ZCA has shown great potential to boost the classification accuracy of deep networks [2], but only when $d$ can be kept small to prevent the training procedure from diverging. To this end, the $c$ output channels of a convolutional layer are partitioned into $G$ groups so that each one contains only $d = c/G$ features. ZCA whitening is then performed within each group independently. This can be understood as a block-diagonal approximation of the complete ZCA whitening. In [2], $d$ is taken to be 3, and the resulting 3×3 covariance matrices are then less likely to have similar or zero eigenvalues. The pseudo-code for ZCA whitening is given in Alg. 1.

We first use CIFAR-10 [20] to compare the behavior of our approach with that of standard SVD and PI for different number of groups $G$, corresponding to different matrix dimensions $d$. Because of numerical instability, the training process of these method often crashes. In short, we observed that

1. For SVD, when $G = 16, d = 4$, 8 out of 15 trials failed; when $G \leq 8, d \geq 8$, the algorithm failed everytime.
2. For PI, when $G = 16, d = 4$, all the trials succeeded; when $G = 8, d = 8$, only 1 out of 15 trials succeeded; when $G \leq 4, d \geq 16$, the algorithm failed everytime.

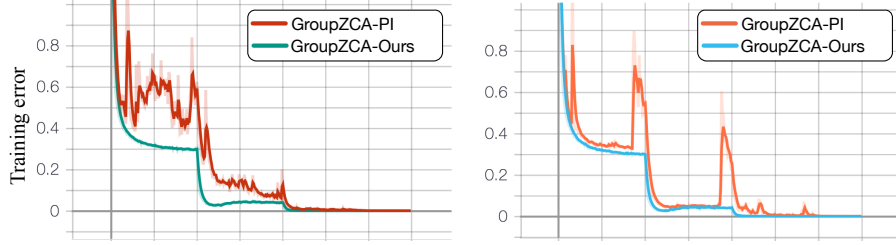

Figure 2: Training loss as a function of the number of epochs for $d=8$ on the left and $d=4$ on the right. In both cases, the loss of standard PI method is very unstable while our method is very stable.

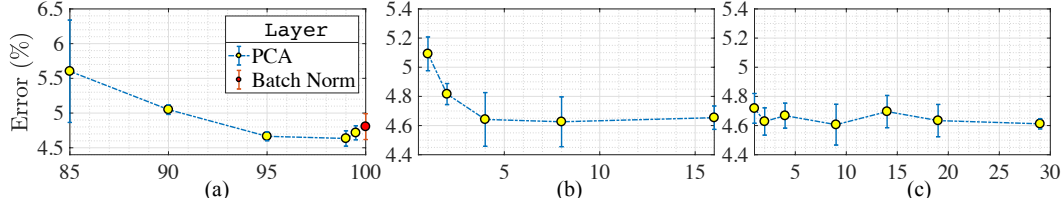

Figure 3: Performance of our PCA denoising layer as a function of (a) the percentage of preserved Information, (b) the number $e$ of eigenvectors retained, (c) the number of power iterations.

3. For our algorithm, we never saw a training failure cases independently of the matrix dimension ranging from $d = 4$ to $d = 64$.

In Table 2, we report the mean classification error rates and their standard deviations over the trials in which they succeeded, along with their success rates. For $d = 4$, SVD unsurprisingly delivers the best accuracy because using analytical derivative is the best thing one can do when possible. However, even for such a small $d$, it succeeds only in $46.7\%$ of trials and systematically fails larger values of $d$. PI can handle $d = 8$ but not consistently as it succeeds only $6.7\%$ of the time. Our approach succeeds for all values of $d$ up to at least 64 and delivers the smallest error rate for $d = 16$, which confirms that increasing the size of the groups can boost performance.

We report equivalent results in Table 3 on CIFAR-100 using either ResNet18 or ResNet50 as the backbone. Our approach again systematically delivers a result. Being able to boost $d$ to 64 for ResNet18 and to 32 for ResNet50 allows us to outperform batch normalization in terms of mean error in both cases.

In Fig. 2, we plot the training losses when using either PI or our method as a function of the number of epochs on both datasets. Note how much smoother ours are.

### 3.2   PCA Denoising

We now turn to PCA denoising that computes the $c \times c$ covariance matrix $\mathbf{M}$ of a $c \times n$ matrix $\mathbf{X}$, finds the eigenvectors of $\mathbf{M}$, projects $\mathbf{X}$ onto the subspace spanned by the $e$ eigenvectors associated to the $e$ largest eigenvalues, with $e < n$, and projects the result back to the original space. Before doing PCA, the matrix $\mathbf{X}$ needs to be standardized by removing the mean and scaling the data by its standard deviation, i.e., $\mathbf{X} = (\mathbf{X}-\mu)/\sigma$.

During training $e$ can be either taken as a fixed number, or set dynamically for each individual batch. In this case, a standard heuristic is to choose it so that $\gamma_e = \sum_{i=1}^{e} \lambda_i / \sum_{i=1}^{n} \lambda_i \geq 0.95$, where the $\lambda_i$ are the eigenvalues, meaning that $95\%$ of the variance present in the original data is preserved. We will discuss both approaches below. As before, we run our training scheme 5 times for each setting of the parameters and report the mean classification accuracy and its variance on CIFAR-10.

**Percentage of Preserved Information vs Performance.**   We first set $e$ to preserve a fixed percentage in each batch as described above. In practice, this means that $e$ is always much smaller than the channel number $c$. For instance, after the first convolutional layer that has 64 channels, we observed that $e$ remains smaller than 7 most of the time when preserving $85\%$ of the variance, 15 when preserving $99\%$, and 31 when preserving $99.5\%$. As can be seen in Fig.3(a) and Table 4(a), retaining less than $90\%$ of the variance hurts performance, and the best results are obtained for $99\%$,

| (a) Preserved Information | | (b) Eigenvector No. | | (c) Power Iteration No. | |
|---|---|---|---|---|---|
| Percentage (%) | Error (%) | Eig-vector No. | Error (%) | Power Iteration No. | Error (%) |
| 85 | $5.60 \pm 0.74$ | 2 | $5.09 \pm 0.12$ | 1 | $4.72 \pm 0.10$ |
| 90 | $5.05 \pm 0.07$ | 4 | $4.82 \pm 0.07$ | 2 | $4.63 \pm 0.09$ |
| 95 | $4.67 \pm 0.06$ | 8 | $4.64 \pm 0.18$ | 4 | $4.67 \pm 0.09$ |
| 99 | $\mathbf{4.63 \pm 0.11}$ | 16 | $\mathbf{4.63 \pm 0.17}$ | 9 | $4.61 \pm 0.14$ |
| 99.5 | $4.72 \pm 0.10$ | 32 | $4.65 \pm 0.08$ | 14 | $4.69 \pm 0.11$ |
| 100 | $4.81 \pm 0.19$ | | | 19 | $4.63 \pm 0.11$ |
| | | | | 29 | $\mathbf{4.61 \pm 0.03}$ |

Table 4: Performance of PCA denoising layer vs Preserved information, Eigenvector No, and PI Number.

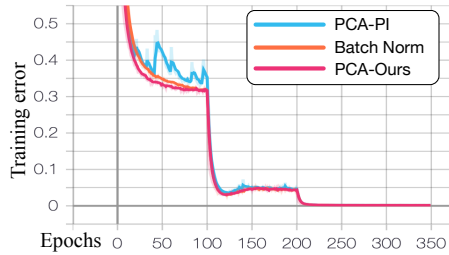

| Methods | Error | Value |
|---|---|---|
| BN | Min | 4.66 |
| | Mean | 4.81±0.19 |
| PCA(PI) | Min | 5.05 |
| | Mean | 5.35±0.25 |
| PCA(SVD) | Min | NaN |
| | Mean | NaN |
| PCA(Ours) | Min | **4.58** |
| | Mean | **4.63±0.11** |

Figure 4: Training loss as a function of the number of epochs.

Table 5: Final error rate for all four methods we tried.

with a mean error of 4.63. Our PCA denoising layer then outperforms Batch Normalization (BN) whose mean error is 4.81, as shown in Table 5.

**Number of Eigenvectors vs Performance.** We now set $e$ to a fixed value. As shown in Fig.3(b) and Table 4(b), the performance is stable when $8 \leq e \leq 32$, and the optimum is reached for $e = 16$, which preserves approximately 99.9% of the variance on average. Note that the average accuracy is the same as in the previous scenario but that its standard deviation is a bit larger.

**Number of Power Iterations vs Performance.** As preserving 99.9% of information yields good results, we now dynamically set $e$ accordingly, and report errors as a function of the number of power iterations during backpropagation in Table 4(c). Note that for our PCA denoising layer, 2 power iterations are enough to deliver an error rate that is very close the best one obtained after 29 iterations, that is, 4.63% of 4.61%. In this case, our method only incurs a small time penalty at run-time. In practice, on one single Titan XP GPU server, for one minibatch with batchsize 128, using ResNet18 as backbone, 2 power iterations take 104.8 ms vs 82.7ms for batch normalization. Note that we implemented our method in Pytorch [21], with the backward pass written in python, which leaves room for improvement.

**Training Stability.** Fig.4 depicts the training curve for our method, PI, and standard BN. Note that the loss values for our method and BN decrease smoothly, which indicates that the training process is very stable. By contrast, the PI training curve denotes far more instability and ultimately lower performance. In Table. 5, we report the final error rates, where the NaN values associated to SVD denotes the fact that using the analytical SVD derivatives failed all five times we tried.

## 4 Discussion & Conclusion

In this paper, we have introduced a numerically stable differentiable eigendecomposition method that relies on the SVD during the forward pass and on Power Iterations to compute the gradients during the backward pass. Both the theory and the experimental results confirm the increased stability that our method brings compared with standard SVD or Power Iterations alone. In addition to performing ZCA more effectively than before, this has enabled us to introduce a PCA denoising layer, which has proven to be effective to improve the performance on classification tasks.

The main limitation of our method is that the accuracy of our algorithm depends on the accuracy of the SVD in the forward pass, which is not always perfect. Besides, the truncation operation in Alg. 1 prevents our method from computing the gradients for the full rank eigenvectors. Moreover, the for-loop in Alg. 1 slows down the speed of the gradient computation. In future work, we will therefore explore approaches to improve it.

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
