[Supplementary Material]

Appendix

## 5   Approximate ED Gradients with PI in Backpropogation

In the following two subsections, we prove that the gradients computed from the PI equals those computed from ED.

### 5.1   Power Iteration Gradients

To compute the leading eigenvector $\mathbf{v}$ of $\mathbf{M}$, PI uses the following standard formula

$$\mathbf{v}^{(k)} = \frac{\mathbf{M}\mathbf{v}^{(k-1)}}{\|\mathbf{M}\mathbf{v}^{(k-1)}\|},\tag{15}$$

where $\|\cdot\|$ denotes the $\ell_2$ norm, and $\mathbf{v}^{(0)}$ is usually initialized randomly with $\|\mathbf{v}^{(0)}\|=1$. Its gradient is [18]

$$\begin{aligned}\frac{\partial L}{\partial \mathbf{M}} &= \sum_k \frac{\left(\mathbf{I}-\mathbf{v}^{(k+1)}\mathbf{v}^{(k+1)\top}\right)}{\|\mathbf{M}\mathbf{v}^{(k)}\|}\frac{\partial L}{\partial \mathbf{v}^{(k+1)}}\mathbf{v}^{(k)\top}\\ \frac{\partial L}{\partial \mathbf{v}^{(k)}} &= \mathbf{M}\frac{\left(\mathbf{I}-\mathbf{v}^{(k+1)}\mathbf{v}^{(k+1)\top}\right)}{\|\mathbf{M}\mathbf{v}^{(k)}\|}\frac{\partial L}{\partial \mathbf{v}^{(k+1)}}\end{aligned}\tag{16}$$

Using 3 power iteration steps for demonstration, we have

$$\begin{aligned}\frac{\partial L}{\partial \mathbf{v}^{(2)}} &= \mathbf{M}\frac{\left(\mathbf{I}-\mathbf{v}^{(3)}\mathbf{v}^{(3)\top}\right)}{\|\mathbf{M}\mathbf{v}^{(2)}\|}\frac{\partial L}{\partial \mathbf{v}^{(3)}}\\ \frac{\partial L}{\partial \mathbf{v}^{(1)}} &= \mathbf{M}\frac{\left(\mathbf{I}-\mathbf{v}^{(2)}\mathbf{v}^{(2)\top}\right)}{\|\mathbf{M}\mathbf{v}^{(1)}\|}\frac{\partial L}{\partial \mathbf{v}^{(2)}} = \mathbf{M}\frac{\left(\mathbf{I}-\mathbf{v}^{(2)}\mathbf{v}^{(2)\top}\right)}{\|\mathbf{M}\mathbf{v}^{(1)}\|}\mathbf{M}\frac{\left(\mathbf{I}-\mathbf{v}^{(3)}\mathbf{v}^{(3)\top}\right)}{\|\mathbf{M}\mathbf{v}^{(2)}\|}\frac{\partial L}{\partial \mathbf{v}^{(3)}}\end{aligned}\tag{17}$$

Then, because we use ED's result, denoted as $\mathbf{v}$, as initial vector, $\mathbf{v}=\mathbf{v}^{(0)}\approx\mathbf{v}^{(1)}\approx\mathbf{v}^{(2)}\approx\cdots\approx\mathbf{v}^{(k)}$. Therefore, $\frac{\partial L}{\partial \mathbf{M}}$ can be re-written as

$$\begin{aligned}\frac{\partial L}{\partial \mathbf{M}} &= \frac{\left(\mathbf{I}-\mathbf{v}^{(3)}\mathbf{v}^{(3)\top}\right)}{\|\mathbf{M}\mathbf{v}^{(2)}\|}\frac{\partial L}{\partial \mathbf{v}^{(3)}}\mathbf{v}^{(2)\top}+\frac{\left(\mathbf{I}-\mathbf{v}^{(2)}\mathbf{v}^{(2)\top}\right)}{\|\mathbf{M}\mathbf{v}^{(1)}\|}\frac{\partial L}{\partial \mathbf{v}^{(2)}}\mathbf{v}^{(1)\top}+\frac{\left(\mathbf{I}-\mathbf{v}^{(1)}\mathbf{v}^{(1)\top}\right)}{\|\mathbf{M}\mathbf{v}^{(0)}\|}\frac{\partial L}{\partial \mathbf{v}^{(1)}}\mathbf{v}^{(0)\top}\\ &= \left(\frac{\left(\mathbf{I}-\mathbf{v}\mathbf{v}^\top\right)}{\|\mathbf{M}\mathbf{v}\|}+\frac{\left(\mathbf{I}-\mathbf{v}\mathbf{v}^\top\right)\mathbf{M}\left(\mathbf{I}-\mathbf{v}\mathbf{v}^\top\right)}{\|\mathbf{M}\mathbf{v}\|^2}+\frac{\left(\mathbf{I}-\mathbf{v}\mathbf{v}^\top\right)\mathbf{M}\left(\mathbf{I}-\mathbf{v}\mathbf{v}^\top\right)\mathbf{M}\left(\mathbf{I}-\mathbf{v}\mathbf{v}^\top\right)}{\|\mathbf{M}\mathbf{v}\|^3}\right)\frac{\partial L}{\partial \mathbf{v}^{(3)}}\mathbf{v}^\top\end{aligned}\tag{18}$$

Since $\mathbf{v}\mathbf{v}^\top$ and $\mathbf{M}$ are symmetric, and $\mathbf{M}\mathbf{v} = \lambda\mathbf{v}$, we have
$$\mathbf{v}\mathbf{v}^\top\mathbf{M} = (\mathbf{M}^\top\mathbf{v}\mathbf{v}^\top)^\top = (\mathbf{M}\mathbf{v}\mathbf{v}^\top)^\top = (\lambda\mathbf{v}\mathbf{v}^\top)^\top = \lambda\mathbf{v}\mathbf{v}^\top = \mathbf{M}\mathbf{v}\mathbf{v}^\top.$$
Introducing the equation above into the numerator of the second term of Eq. 18 yields
$$\begin{aligned}\left(\mathbf{I}-\mathbf{v}\mathbf{v}^\top\right)\mathbf{M}\left(\mathbf{I}-\mathbf{v}\mathbf{v}^\top\right) &= \left(\mathbf{M}-\mathbf{v}\mathbf{v}^\top\mathbf{M}\right)\left(\mathbf{I}-\mathbf{v}\mathbf{v}^\top\right) = \left(\mathbf{M}-\mathbf{M}\mathbf{v}\mathbf{v}^\top\right)\left(\mathbf{I}-\mathbf{v}\mathbf{v}^\top\right)\\ &= \mathbf{M}\left(\mathbf{I}-\mathbf{v}\mathbf{v}^\top\right)\left(\mathbf{I}-\mathbf{v}\mathbf{v}^\top\right) = \mathbf{M}\left(\mathbf{I}-2\mathbf{v}\mathbf{v}^\top+\mathbf{v}(\mathbf{v}^\top\mathbf{v})\mathbf{v}^\top\right) = \mathbf{M}\left(\mathbf{I}-\mathbf{v}\mathbf{v}^\top\right).\end{aligned}\tag{19}$$
Similarly, for the numerator in the third term in Eq.18, we have
$$\left(\mathbf{I}-\mathbf{v}\mathbf{v}^\top\right)\mathbf{M}\left(\mathbf{I}-\mathbf{v}\mathbf{v}^\top\right)\mathbf{M}\left(\mathbf{I}-\mathbf{v}\mathbf{v}^\top\right) = \mathbf{M}\mathbf{M}\left(\mathbf{I}-\mathbf{v}\mathbf{v}^\top\right).\tag{20}$$
Introducing Eq.19 and Eq.20 into Eq.18, we obtain
$$\frac{\partial L}{\partial \mathbf{M}} = \left(\frac{\left(\mathbf{I}-\mathbf{v}\mathbf{v}^\top\right)}{\|\mathbf{M}\mathbf{v}\|}+\frac{\mathbf{M}\left(\mathbf{I}-\mathbf{v}\mathbf{v}^\top\right)}{\|\mathbf{M}\mathbf{v}\|^2}+\frac{\mathbf{M}\mathbf{M}\left(\mathbf{I}-\mathbf{v}\mathbf{v}^\top\right)}{\|\mathbf{M}\mathbf{v}\|^3}\right)\frac{\partial L}{\partial \mathbf{v}^{(3)}}\mathbf{v}^\top\tag{21}$$

When extending the iteration number from 3 to $k$, Eq.18 becomes
$$\frac{\partial L}{\partial \mathbf{M}} = \left(\frac{\left(\mathbf{I}-\mathbf{v}\mathbf{v}^\top\right)}{\|\mathbf{M}\mathbf{v}\|}+\frac{\mathbf{M}\left(\mathbf{I}-\mathbf{v}\mathbf{v}^\top\right)}{\|\mathbf{M}\mathbf{v}\|^2}+\cdots+\frac{\mathbf{M}^{k-1}\left(\mathbf{I}-\mathbf{v}\mathbf{v}^\top\right)}{\|\mathbf{M}\mathbf{v}\|^k}\right)\frac{\partial L}{\partial \mathbf{v}^{(k)}}\mathbf{v}^\top\tag{22}$$

Eq.22 is the form we adopt to compute the gradients of ED.

## 5.2 Analytic ED Gradients

The analytic solution of the ED gradients is [4].

$$\frac{\partial L}{\partial \mathbf{M}} = V \left\{ \left( \tilde{K}^\top \circ \left( V^\top \frac{\partial L}{\partial V} \right) \right) + \left( \frac{\partial L}{\partial \Sigma} \right)_{diag} \right\} V^\top \tag{23}$$

$$\tilde{K}_{ij} = \begin{cases} \frac{1}{\lambda_i - \lambda_j}, & i \neq j \\ 0, & i = j \end{cases} \tag{24}$$

$$\tilde{K} = \begin{bmatrix} 0 & \frac{1}{\lambda_1 - \lambda_2} & \frac{1}{\lambda_1 - \lambda_3} & \cdots & \frac{1}{\lambda_1 - \lambda_n} \\ \frac{1}{\lambda_2 - \lambda_1} & 0 & \frac{1}{\lambda_2 - \lambda_3} & \cdots & \frac{1}{\lambda_2 - \lambda_n} \\ \frac{1}{\lambda_3 - \lambda_1} & \frac{1}{\lambda_3 - \lambda_2} & 0 & \cdots & \frac{1}{\lambda_3 - \lambda_n} \\ \vdots & \vdots & \vdots & \ddots & \vdots \\ \frac{1}{\lambda_n - \lambda_1} & \frac{1}{\lambda_n - \lambda_2} & \frac{1}{\lambda_n - \lambda_3} & \cdots & 0 \end{bmatrix} \tag{25}$$

where $\lambda_i$ is an eigenvalue, and

$$V = \begin{bmatrix} \mathbf{v}_1 & \mathbf{v}_2 & \mathbf{v}_3 & \cdots & \mathbf{v}_n \end{bmatrix} \tag{26}$$

where $\mathbf{v}_i$ is an eigenvector. Then,

$$\frac{\partial L}{\partial V} = \begin{bmatrix} \frac{\partial L}{\partial \mathbf{v}_1} & \frac{\partial L}{\partial \mathbf{v}_2} & \frac{\partial L}{\partial \mathbf{v}_3} & \cdots & \frac{\partial L}{\partial \mathbf{v}_n} \end{bmatrix} \tag{27}$$

$$V^\top \frac{\partial L}{\partial V} = \begin{bmatrix} \mathbf{v}_1^\top \frac{\partial L}{\partial \mathbf{v}_1} & \mathbf{v}_1^\top \frac{\partial L}{\partial \mathbf{v}_2} & \mathbf{v}_1^\top \frac{\partial L}{\partial \mathbf{v}_3} & \cdots & \mathbf{v}_1^\top \frac{\partial L}{\partial \mathbf{v}_n} \\ \mathbf{v}_2^\top \frac{\partial L}{\partial \mathbf{v}_1} & \mathbf{v}_2^\top \frac{\partial L}{\partial \mathbf{v}_2} & \mathbf{v}_2^\top \frac{\partial L}{\partial \mathbf{v}_3} & \cdots & \mathbf{v}_2^\top \frac{\partial L}{\partial \mathbf{v}_n} \\ \mathbf{v}_3^\top \frac{\partial L}{\partial \mathbf{v}_1} & \mathbf{v}_3^\top \frac{\partial L}{\partial \mathbf{v}_2} & \mathbf{v}_3^\top \frac{\partial L}{\partial \mathbf{v}_3} & \cdots & \mathbf{v}_3^\top \frac{\partial L}{\partial \mathbf{v}_n} \\ \vdots & \vdots & \vdots & \ddots & \vdots \\ \mathbf{v}_n^\top \frac{\partial L}{\partial \mathbf{v}_1} & \mathbf{v}_n^\top \frac{\partial L}{\partial \mathbf{v}_2} & \mathbf{v}_n^\top \frac{\partial L}{\partial \mathbf{v}_3} & \cdots & \mathbf{v}_n^\top \frac{\partial L}{\partial \mathbf{v}_n} \end{bmatrix} \tag{28}$$

$$\tilde{K} \circ V^\top \frac{\partial L}{\partial V} = \begin{bmatrix} 0 & \frac{1}{\lambda_2 - \lambda_1} \mathbf{v}_1^\top \frac{\partial L}{\partial \mathbf{v}_2} & \frac{1}{\lambda_3 - \lambda_1} \mathbf{v}_1^\top \frac{\partial L}{\partial \mathbf{v}_3} & \cdots & \frac{1}{\lambda_n - \lambda_1} \mathbf{v}_1^\top \frac{\partial L}{\partial \mathbf{v}_n} \\ \frac{1}{\lambda_1 - \lambda_2} \mathbf{v}_2^\top \frac{\partial L}{\partial \mathbf{v}_1} & 0 & \frac{1}{\lambda_3 - \lambda_2} \mathbf{v}_2^\top \frac{\partial L}{\partial \mathbf{v}_3} & \cdots & \frac{1}{\lambda_n - \lambda_2} \mathbf{v}_2^\top \frac{\partial L}{\partial \mathbf{v}_n} \\ \frac{1}{\lambda_1 - \lambda_3} \mathbf{v}_3^\top \frac{\partial L}{\partial \mathbf{v}_1} & \frac{1}{\lambda_2 - \lambda_3} \mathbf{v}_3^\top \frac{\partial L}{\partial \mathbf{v}_2} & 0 & \cdots & \frac{1}{\lambda_n - \lambda_3} \mathbf{v}_3^\top \frac{\partial L}{\partial \mathbf{v}_n} \\ \vdots & \vdots & \vdots & \ddots & \vdots \\ \frac{1}{\lambda_1 - \lambda_n} \mathbf{v}_n^\top \frac{\partial L}{\partial \mathbf{v}_1} & \frac{1}{\lambda_2 - \lambda_n} \mathbf{v}_n^\top \frac{\partial L}{\partial \mathbf{v}_2} & \frac{1}{\lambda_3 - \lambda_n} \mathbf{v}_n^\top \frac{\partial L}{\partial \mathbf{v}_3} & \cdots & 0 \end{bmatrix} \tag{29}$$

$$V \tilde{K} \circ V^\top \frac{\partial L}{\partial V} = \begin{bmatrix} \sum_{i \neq 1}^n \frac{1}{\lambda_1 - \lambda_i} \mathbf{v}_i \mathbf{v}_i^\top \frac{\partial L}{\partial \mathbf{v}_1}, & \cdots, & \sum_{i \neq n}^n \frac{1}{\lambda_n - \lambda_i} \mathbf{v}_i \mathbf{v}_i^\top \frac{\partial L}{\partial \mathbf{v}_n} \end{bmatrix} \tag{30}$$

$$V \tilde{K} \circ V^\top \frac{\partial L}{\partial V} V^\top = \sum_{i \neq 1}^n \frac{1}{\lambda_1 - \lambda_i} \mathbf{v}_i \mathbf{v}_i^\top \frac{\partial L}{\partial \mathbf{v}_1} \mathbf{v}_1 + \cdots + \sum_{i \neq n}^n \frac{1}{\lambda_n - \lambda_i} \mathbf{v}_i \mathbf{v}_i^\top \frac{\partial L}{\partial \mathbf{v}_n} \mathbf{v}_n \tag{31}$$

$$V \left( \frac{\partial L}{\partial \Sigma} \right)_{diag} V^\top = \sum_{i=1}^n \frac{\partial L}{\partial \lambda_i} \mathbf{v}_i \mathbf{v}_i^\top \tag{32}$$

Let us now consider the partial derivative *w.r.t.* the dominant eigenvector $\mathbf{v}_i$ and ignore the remaining $\frac{\partial L}{\partial \mathbf{v}_i}, i \neq 1$. Then $\frac{\partial L}{\partial \mathbf{M}}$ becomes

$$\frac{\partial L}{\partial M} = \sum_{i=2}^n \frac{1}{\lambda_1 - \lambda_i} \mathbf{v}_i \mathbf{v}_i^\top \frac{\partial L}{\partial \mathbf{v}_1} \mathbf{v}_1^\top + \frac{\partial L}{\partial \lambda_1} \mathbf{v}_1 \mathbf{v}_1^\top . \tag{33}$$