[Reviews · NeurIPS 2019]

Reviewer 1



* First, for a nonsymmetric matrix, the eigenvalues are not related to the SVD by a variational principle. This is a point of confusion throughout the paper. In line 57, you start with `given an input matrix M'. Listing the SVD / QR as a method to compute the the ED is not correct here. Later you clarify that M is considered to be a covariance matrix. You should clarify at the very beginning that the scope of your method is limited and restricted to symmetric matrices! * It is not clear to me why it is preferable to use power iterations for approximating the gradients. You are saying in line 28 that the power iteration method is numerically not stable if two eigenvalues being close. So what is the advantage of using power iterations compared to using the analytic solution for the gradients? * Well, you introduce some tricks like ridge regularization later in order to stabilize your computations. But what if you set your epsilon in line 117 to 0? I assume you would gain not much... * This said, your computational experiments are a bit biased, in a sense that you have introduced a hidden regularization parameter in your network, namely epsilon. I assume that setting epsilon to 10^-4 has quite some effect. What happens if you set epsilon to 10^-12. Further, you compare your method, that uses several practical tricks (Sec 2.5), to a plain implementation of the SVD. This comparison does not seem to be fair, since you can, for instance, also truncate the ordinary SVD. Thus, I think that a fair comparison requires the use of the same practical tricks. * I am not very impressed by your results. I do not see much benefit of using ZCA or PCA denoising here, in particular not for CIFAR10. Your results for CIFAR100 are somewhat far away from current state-of-the-art performance (correct me if I am wrong here!). Further it would be nice to compare results to [r1]. * Is PCA denoising really a new normalizing strategy for deep network? * Overall, the quality of the writing is good. Smaller comments: * I doubt that the eigendecomposition is widely used in deep networks. That is, because there are so many numerical issues. You may want to modify the abstract accordingly. * You should introduce L in Eq. (3). * Why is Sigma in Eq. (5) not bold? [r1] Iterative Normalization: Beyond Standardization towards Efficient Whitening. CVPR. 2019.

Reviewer 2



The paper proposes a numerically stable and back propagation compatible eigendecomposition for deep neural networks. It addresses the instability issue in analytic derivatives, as well as the convergence issue for power iterations. The authors give the theoretical justification behind their approach, while showing the numerical evidence behind the choice of parameters. The algorithm is robust when applied in ZCA and PCA denoising, which marginally improved the performance deep networks on CIFAR_10/100 and ResNet18/50. The paper is sound, though I am not familiar enough to comment on the originality. The algorithm should be useful in practice for handling the corner cases. Major comments: 1. The author mentioned a few times in the paper that the method is intended for large matrices, yet most matrices in the experiments have relatively small size. I am curious if the failure cases become more or less common, when the matrices become large enough. Minor comments: 1. The improvement in the experiments seem rather small. Most of the time, the convergence behavior looks very similar whether using the existing methods or the proposed one. Post author feedback: Thank the authors for addressing my questions. I think the problem this paper is trying to tackle is important. Unfortunately, I am still not convinced whether using a power series approximation to the derivative when the eigendecomposition is not differentiable is the best approach in this case, and whether it leads to meaningful improvement in applications. As I am remain mostly neutral on this paper, I decide to leave my score unchanged.

Reviewer 3



The basic idea is very simple: for a PCA layer in a DNN, both SVD and power iteration methods have drawbacks. To overcome this, we run SVD in the forward pass and power iteration in the backward pass. This leads to an "improper" backprop, but the results are close enough (people do this all the time, for example in batch renorm) or the errors introduced are good enough that it doesn't matter. I think that originality is somewhat limited in this paper (i.e., it combines two well-known elements in a well-known way). But I do think that the superiority of this approach to previous ones speaks to its value. The experimental evaluation of the main method is sound and convincing. The crucial experiment for me is Table 2 that shows that basically d = 64 works for this method, whereas previous methods could only really handle d=4. A very natural question here is whether we even have to do blocks of d. On the other hand, I'm not totally convinced about Sec 3.2 and the PCA denoising: realistically, it doesn't look any better than batch normalization. I think perhaps trying this on non-residual networks might give better results, or comparing to vanilla no BN networks. The paper is well written. Even in the face of quite a lot of mathematics, the paper is clear and a pleasure to read. I found only one typo (line 106 'backpropogation'). This is a high quality submission, and the authors have obviously put effort into the writing: thank you! The significance of this paper is hurt a little by the niche nature of PCA layers, but as above, I think that this paper could be the basis of a lot of new ideas, so overall I think significance is high. (It's not clear to me that citing Eigenfaces as an application of PCA layers to deep learning is appropriate?). POST AUTHOR FEEDBACK: Thank you for the the clarifications. My critiques were quite minor before, so nothing significant has changed in the score, leaving as is.

[Author Response · NeurIPS 2019]

**R1: For a nonsymmetric matrix ...** We will clarify from the start that our method is designed for symmetric matrices.
**R1: PI is unstable if two eigenvalues are close. So what is the advantage of PI vs the analytic solution?** The
instability of PI is due to inaccurate initialization of the eigenvectors. In our case, the forward pass provides accurate
values using SVD and PI then becomes stable during the backward pass. Another advantage over the analytic solution
what we demonstrate is that **PI yields an upper bound on the gradients' magnitude, which guarantees they will not
explode.** As shown in Sec. 2.2, PI is the geometric expansion of the analytic solution: With $\lambda_i \geq \lambda_j$, the term $1/(\lambda_i - \lambda_j)$,
which appears in the analytic solution, is approximated by $1/\lambda_i + 1/\lambda_i (\lambda_j/\lambda_i) + 1/\lambda_i (\lambda_j/\lambda_i)^2 + \cdots + 1/\lambda_i (\lambda_j/\lambda_i)^{K-1} \leq$
$K/\lambda_i$ in PI, where $K$ represents the power iteration number. When $\lambda_i = \lambda_j$, the term $1/(\lambda_i - \lambda_j)$ in the analytic solution
explodes, while the PI gradients are naturally upper bounded by $K/\lambda_i$.
**R1: A hidden regularizer, $\epsilon$, is introduced. What if you set $\epsilon$ to $0$ or $10^{-12}$.** $\epsilon$ is commonly used to stabilize the
computation (*e.g.,* [2] [3] & [r1]). As shown in Eq. 13, $\epsilon$ controls the gradients' upper bound and appears in the
denominator. Thus, too small an epsilon, *e.g.,* $10^{-12}$, will yield a large upper bound and increase the risk of gradient
explosion. Following standard practice, e.g., [3], $\epsilon$ is set to $10^{-4}$ for *all* the methods, including SVD and PI, but our
method always achieves 100% success rate while the others do not.
**R1: Your method uses some tricks. The comparison seems unfair, since you can also truncate SVD.** For a fair
comparison, we recomputed results using the same tricks to truncate the eigenvalues for our SVD baseline. For PI,
truncation was already used in our submission to mitigate the round-off errors caused by the deflation process. The
results on CIFAR10 are given in the table below. Note that, for matrix dimensions $d > 16$, SVD and PI still fail in all
cases. By contrast, we achieve 100% success rate even when the dimension is as large as 128.

| Methods | Evaluation Metrics | $d = 4$ | $d = 8$ | $d = 16$ | $d = 32$ | $d = 64$ | $d = 128$ |
|---|---|---|---|---|---|---|---|
| SVD | *Min Error & Suc. Rate* | 4.50%, 60% | 4.75%, 33.3% | 4.65%, 40% | $-$, 0% | $-$, 0% | $-$, 0% |
| PI | *Min Error & Suc. Rate* | 4.44%, **100**% | 6.28%, 6.7% | $-$, 0% | $-$, 0% | $-$, 0% | $-$, 0% |
| Ours | *Min Error & Suc. Rate* | 4.59%, **100**% | 4.43%, **100**% | **4.40**%, **100**% | **4.46**%, **100**% | **4.44**%, **100**% | **4.75**%, **100**% |

**R1: Results on CIFAR100 are far away from SOTA performance. It would be nice to compare with [r1].** Our
paper focuses on solving the stability issues of ED, not designing better normalization layers (*i.e.*, PCA & ZCA).
Stability is measured as the success rate of the methods, which, for our purpose, is more important than accuracy. ZCA
and PCA constitute two applications of our method to demonstrate stability. We therefore just used simple backbones
(*i.e.*, ResNet18/50), which translates to accuracies inferior to the SOTA. By contrast, [r1] focuses on designing a better
normalization layer using an iterative normalization method. We nonetheless acknowledge the relevance of this paper,
which we will cite in the final version. Note that [r1] was not published at the time of NeurIPS submission.
**R1: I doubt ED is widely used as there are numerical issues. Justify why ED is important for deep learning.**
Indeed, ED has many numerical stability issues, and this is exactly what our paper addresses. Nevertheless, as discussed
in the introduction from Line 14 to 18, ED has been used for image/point matching [6,7,8], second-oder pooling [4],
and pose estimation [9]. It has not been well integrated into deep networks because of the numerical instability, and, as
stated by R3, one can expect that our paper will bring insight to this problem and be the basis for many new ideas.

**R2: Will the failure cases become more or less common when matrices are large enough?** As shown in the table
above, the baselines' failures become more common as $d$ increases, whereas our method succeeds 100% of the time for
all dimensions. This remains true when increasing $d$ to 128 (twice as many as in the submission), by putting the ZCA
layer on top of a 128-channel conv. layer. The underlying reasons are that, thanks to our use of SVD in the forward pass,
we have more accurate eigen value/vector estimates than the PI baseline, and that, as shown in Eq. 13, the gradients of
our method are always bounded regardless of matrix size while those of the SVD baseline may easily explode.
**R2: The convergence behavior looks similar whether using the existing methods and the proposed one.** The
convergence curves shown in Fig. 2 are based solely on the successful cases for the baselines and ignore the failure cases
(see success rates in Table 2). Including these numerous failures would render the baseline curves entirely meaningless.
**R2: In Tables 3 and 4, the prediction error is not monotone with the matrix size.** For ZCA in Table 3, with
ResNet18, all values are virtually the same, and with ResNet50, the trend shows that larger $d$ values, which only our
method can handle, give better results. For PCA in Table 4 (a,b), when too much information is preserved, some noise
is kept and the accuracy drops. Conversely, when too much information is removed, some useful signal is discarded and
the accuracy also drops. The right number of dimensions to preserve is dataset dependent and can be determined by
cross validation.
**R3: Do we have to do blocks of $d$?** Dividing the features into blocks of dimension $d$ is only useful to compare our
approach with the baselines, which only succeed for small matrix dimensions. Given our stabilized ED method, the
blocks become unnecessary, and the largest dimension $d$ in each experiment corresponds to not using blocks.
**R3: PCA denoising doesn't look better than batch normalization (BN).** While PCA denoising indeed has
marginal improvement over BN, it is not our main focus. Similarly to ZCA, PCA denoising only is another ap-
plication to demonstrate the stability of our method. The baselines to truly look at in Table 5 are PCA(PI) and
PCA(SVD), which often break down in the training phase. We will emphasize this in the final version.
**R3: Is there a way of minimizing the dependency on ZCA?** We will minimize the emphasis on ZCA whitening
and PCA denoising in the abstract and introduction.

[Meta-Review · NeurIPS 2019]

There is mixed feedback in the reviews on the significance of the paper. In my view it is a fundamental building block in deep learning to be able to include modules that perform eigendecompositions in a stable and differentiable (via backprop) manner. While the idea of the paper is simple, i.e. combine the advantages of SVD and power iterations by using them in the forward and backward pass, respectively, there is real insight (see stability bounds) and solid empirical evidence (100% stability even in relatively high dimensions). I think this clarity - simple, but important idea, well analyzed - is a feature and makes this paper a valuable contribution.